# Combined abdominal heterotopic heart and aorta transplant model in mice

Hao Dun, Li Ye, Yuehui Zhu, Brian W. Wong *

Laboratory of Lymphatic Metabolism + Epigenetics, Department of Surgery, Washington University School of Medicine, St. Louis, MO, United States of America

* brian.wong@wustl.edu

## Abstract

### Background

Allograft vasculopathy (AV) remains a major obstacle to long-term allograft survival. While the mouse aortic transplantation model has been proven as a useful tool for study of the pathogenesis of AV, simultaneous transplantation of the aorta alongside the transplantation of another organ may reveal more clinically relevant mechanisms that contribute to the pathogenesis of chronic allograft rejection. Therefore, we developed a combined abdominal heart and aorta transplantation model in mice which benefits from reducing animal and drug utilization, while providing an improved model to study the progressive nature of AV.

### Methods

The middle of the infrarenal aorta of the recipient mouse was ligatured between the renal artery and its bifurcation. Proximal and distal aortotomies were performed at this site above and below the ligature, respectively, for the subsequent anastomoses of the donor aorta and heart grafts to the recipient infrarenal aorta in an end-to-side fashion. The distal anastomotic site of the recipient infrarenal aorta was connected with the outlet of the donor aorta. Uniquely, the proximal anastomotic site on the recipient infrarenal aorta was shared to connect with both the inlet of the donor aorta and the inflow tract to the donor heart. The outflow tract from the donor heart was connected to the recipient inferior vena cava (IVC).

### Results

The median times for harvesting the heart graft, aorta graft, recipient preparation and anastomosis were 11.5, 8.0, 9.0 and 40.5 min, respectively, resulting in a total median ischemic time of 70 min. The surgery survival rate was more than 96% (29/30). Both the syngeneic *C57Bl/6* aorta and heart grafts survived more than 90 days in 29 *C57Bl/6* recipients. Further, *Balb/c* to *C57Bl/6* allografts treated with anti-CD40L and CTLA4.Ig survived more than 90 days with a 100% (3/3) survival rate. (3/3).

### Conclusions

This model is presented as a new tool for researchers to investigate transplant immunology and assess immunosuppressive strategies. It is possible to share a common anastomotic

**Data Availability Statement:** All relevant data are within the paper and its Supporting Information files.

**Funding:** This work was supported by the Joel D. Cooper Career Development Award from the

International Society of Heart and Lung Transplantation (BWW). The funders had no role in study design, data collection and analysis, decision to publish, or preparation of the manuscript. There was no additional external funding received for this study.

**Competing interests:** The authors have declared that no competing interests exist.

**Abbreviations:** AV, allograft vasculopathy; CAV, cardiac allograft vasculopathy; IVC, inferior vena cava; MHC, major histocompatibility complex; SVC, superior vena cava.

stoma on the recipient abdominal aorta to reconstruct both the aorta graft entrance and heart graft inflow tract. This allows for the study of allogeneic effects on both the aorta and heart from the same animal in a single survival surgery.

## Introduction

Solid organ transplantation is one of the only viable interventions for patients with end-stage organ failure. Beyond acute rejection, which is most prevalent in the first year post-transplantation and can be actively managed with adjustments in immunosuppressive regimen or induction therapy [1, 2], the long term survival of transplanted hearts is impeded by graft failure, malignancy, cardiac allograft vasculopathy (CAV) and renal failure [1, 2]. Chronic allograft rejection remains one of the leading causes of graft failure one year post-transplantation [1, 2].

It is well known that T cell-mediated immune responses play a central role in acute allograft rejection [3]. Current immunosuppressive regimens targeting T cell activation and effector cell function have led to dramatic reductions in acute rejection. However, chronic allograft injury leading to graft failure and CAV remains a major obstacle to the long-term allograft survival [1, 2, 4]. Although the exact etiology remains unclear, multifactorial mechanisms including both immunological and non-immunological components contribute to the development of chronic allograft rejection [5, 6]. In the heart, chronic allograft rejection presents as CAV, and is characterized as an accelerated form of atherosclerosis which occurs in the arteries of the transplanted heart [5, 6]. CAV is initiated by a combination of ischemia/reperfusion injury and alloimmune injury which results in endothelial dysfunction [5, 6]. This leads to a progressive fibroproliferative disease with intimal smooth muscle cell proliferation leading to progressive vessel occlusion, thrombotic events and eventual graft failure [5, 6]. Allograft vasculopathy (AV) can also occur in a number of solid organ transplant settings (i.e. lung, kidney, etc.) with similar histopathological characteristics to CAV [7]. Despite current immunosuppressive regimens, CAV is reported in almost 50% of patients 10 years post transplantation [1, 2]. As such, there is a growing need for the development of reliable animal models to decipher underlying mechanism of CAV and further optimize and develop therapeutic strategies to address this major health burden.

Heterotopic heart transplantation in mice has been considered the pre-eminent model to study transplant immunology since it was introduced by Corry and Russell in 1973 [8]. Without immunosuppression, transplantation of a fully MHC-mismatched cardiac allograft induces strong alloreactive T cell responses that mediate rapid graft rejection [9]. In this regard, the heterotopic heart transplant model in mice recapitulates the pathological process of acute allograft rejection. To study chronic allograft injury, immunosuppressive drugs have used in this model to suppress the acute immune response. However, the majority of currently used immunosuppressive drugs already target T cell activation to prevent acute transplant rejection. Thus, the development of AV is likely a reflection of sub-acute immunological events. Further, it has been suggested that these immunosuppressive agents may also contribute to the development of AV [10].

The aortic transplant model in mice has been used to study some components of the immunological and/or molecular mechanisms of AV. However, fully MHC-mismatched aortic allografts demonstrate long-term survival even in absence of immunosuppression [11]. On one hand, this allows for the study of mechanisms that contribute to AV in the absence of a strict

influence from acute rejection, as acute rejection episodes have long been considered as a risk factor for the future development of AV [12]. However, as aortic allografts do not undergo acute rejection, it remains controversial whether the vascular changes observed in aortic allografts accurately represent those that occur in solid organ transplants [13].

To address some of these issues, investigators have developed combined heart and aorta/carotid artery transplantation models in mice to investigate the potential impact of acute rejection on CAV [13, 14]. As expected, compared with isolated carotid allografts, a significantly more intimal hyperplasia of carotid allografts was noted in aorta transplanted in combination with a heart graft, indicating that CAV is promoted by acute parenchymal rejection of the heart [14]. This observation has further highlighted the need for transplantation of a parenchymal organ in combination with an aortic graft to more accurately model the events which contribute to the pathogenesis of CAV in human heart allografts. Despite impressive results made utilizing their models, the transplantation of heart and aorta/carotid graft into two different operative sites (abdominal and cervical respectively) causes increased operative traumas and prolonged operative times, limiting its widespread use. In order to simplify the surgical protocol and remove the need for multiple survival surgeries (resulting in reductions in graft ischemia and overall operative times), we developed a new technique to transplant heart and aorta graft in a single site within the abdomen. Further, transplantation of the aorta and heart in a single surgery reduces donor animal utilization. The aim of our study was therefore to evaluate a new microsurgical technique of simultaneous heterotopic abdominal heart and aorta transplantation in mice, as well as to assess the feasibility of sharing a common anastomotic stoma on the recipient infrarenal aorta to reconstruct both the aorta-graft entrance and heart-graft inflow tract.

Our results demonstrate that this new surgical protocol is reliable, reproducible and improves upon existing techniques. Further studies using this model will provide insight into the clinical process of chronic allograft rejection and a useful tool for assessing the novel immunosuppressive strategies for its prevention. In addition, to our knowledge, this is the first demonstration of a novel microsurgical technique in which an anastomotic stoma on the recipient infrarenal aorta can be connected with two individual vessels, providing a new pathway of revascularization for multi-organ transplantation.

## Materials and methods

### Animals

Adult female *Balb/c* mice 6–10 weeks of age (20-25g body weight) and male *C57Bl/6* mice, 6–10 weeks of age (20-25g body weight) were purchased from Jackson Laboratory (Bar Harbor, MD, USA). Animals were housed under standard conditions subjected to regular 12-hour light-dark cycles. Water and chow were supplied *ad libitum*. Animal experiments were conducted in accordance with an approved Washington University School of Medicine (WUSM) Institutional Animal Care and Use Committee (IACUC) protocol (#20190173). Anesthesia was induced by a mixture of ketamine (80–100 mg/kg)/xylazine HCl (8–12 mg/kg), intraperitoneally (i.p.) and maintained with 1–2% isoflurane gas, as required.

### Donor operation

A longitudinal laparotomy was made, and abdominal contents were reflected to the left side to expose the inferior vena cava (IVC). Approximately 1.0 mL of cold saline containing heparin (100 U/mL) was injected into the IVC. After 1 minute for systemic heparinization, an aortotomy of the abdominal aorta was made to decompress the blood circulatory system. Subsequently, a bilateral thoracotomy was performed through the ribs along both sides of the

thoracic spine, then the anterior chest wall was levered up cranially. The IVC around diaphragmatic hiatus was clamped with a hemostat, above that, the IVC was cannulated and then 0.5 mL of cold heparin (100 U/mL) is again infused into the right atrium. The thymus was resected to expose the aortic arch and pulmonary artery. The ascending aorta was transected proximal to the innominate artery. The pulmonary artery was transected proximal to its bifurcation. The IVC and the right superior vena cava (SVC) were proximally ligated with 7–0 sutures. A 5–0 ligature was placed underneath the IVC and the right SVC and around the heart to ligate all other vessels *en bloc* including the pulmonary veins and the left SVC as distally as possible. The IVC, the right SVC and the remaining connective tissues were all dissected distally in order. The donor heart was then gently detached from the thorax and stored in ice-cold saline. In the following steps, the donor thoracic descending aorta would be harvested as described by Cho *et al* [15]. Lungs were resected, then the diaphragm was divided as close to the aorta as possible to expose descending aorta. The descending aorta was further flushed by cold heparin (100 U/mL) solution from the aorta bellow the diaphragm. The parietal pleura in front of the descending aorta were divided. Intercostal arteries were carefully divided from the segment of the descending aorta between the left subclavian artery proximally and the diaphragmatic hiatus distally. The thoracic descending aorta was harvested and stored in ice-cold saline. Although the thoracic aorta can be divided into three segments for individual transplantation, it is preferable to use the proximal section, as it is most compatible in diameter with ascending aorta of the donor heart, and therefore facilitates the subsequent anastomosis.

## Recipient operation

A midline laparotomy was made from the pubis to the xiphoid, then a micro-retractor was placed to expose the abdominal cavity. The intestines were gently retracted right outside the abdomen and covered with moistened gauze and kept moist throughout the procedure with saline. Using two cotton swabs, the abdominal aorta and the IVC below the renal vessels and above its bifurcation were exposed and dissected gently from the surrounding tissues. Of note, a complete isolation of infrarenal aorta from IVC was unnecessary for performing subsequent end-to-side anastomosis. One or two groups of the lumber arteries and veins underneath the abdominal aorta and IVC were ligated with 9–0 silk sutures. To interrupt the blood flow in both the aorta and the IVC, a 5–0 silk was single-tied at the proximal side of its bifurcation to allow it to be untied easily after anastomosis, and then a micro clamp was placed just at below the renal vessels. Next, the middle of the sequestered aorta was sutured with a 9–0 nylon suture (**Fig 1A and 1B**). Thus, the aorta was partitioned into proximal and distal anastomotic areas without transection. The proximal and distal aortotomies were performed above and below the ligature respectively, for the subsequent anastomoses of the aorta- graft into the recipient in an end-to-side fashion (**Fig 2A**). We preferred to make the aortotomy by a simple and reliable technique described by Mao *et al.* [16]. Briefly, the tip of a 4 mm (3/8) needle was longitudinally passed in and slightly out of the anterior wall of the recipient aorta and kept in position by a needle holder. By gently lifting the needle upwardly, the piece of the anterior wall of the aorta above the needle could be easily excised by cutting underneath the needle with fine scissors, so that an approximate 1 mm elliptical opening with the trim-edge was made. In the same way, a venotomy in parallel with proximal aortotomy was made for the subsequent construction of heart-graft outflow tract. The openings of the aorta and the IVC were flushed with saline to remove the blood.

The aorta graft was placed on the left side of the recipient abdomen, and then covered with gauzes moistened with icy cold saline. First, using 11–0 nylon surgical suture, the aorta-graft exit was constructed in the distal anastomotic site of the recipient infrarenal aorta in an end-

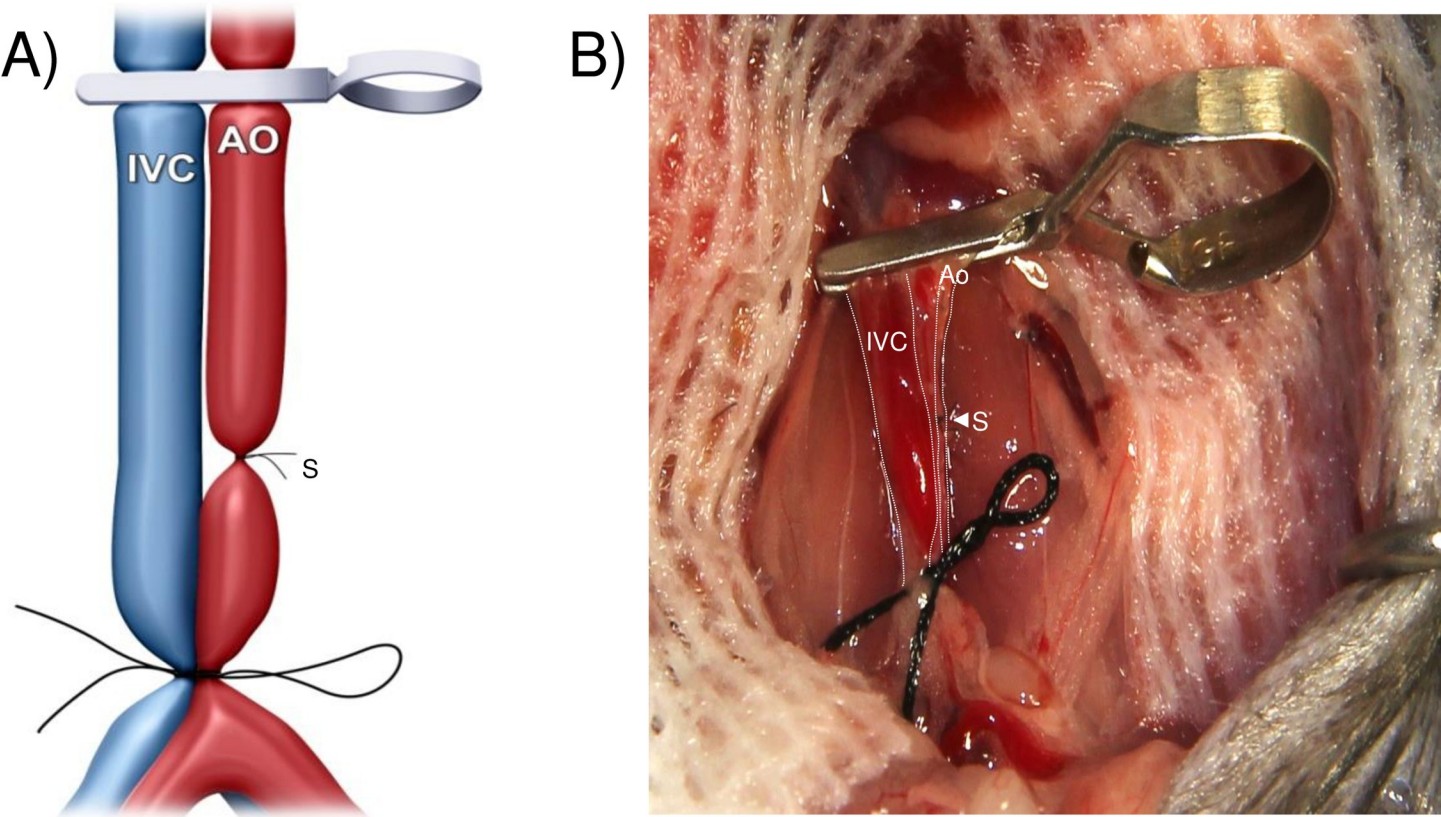

**Fig 1.** Diagrammatic (A) and photographic (B) representation of the middle of the recipient's infrarenal aorta between renal artery and its bifurcation, which was ligatured with a 9–0 nylon suture.

to-side fashion. The posterior and the anterior walls of the anastomosis were completed inside and outside the vessels with 4–5 continuous running sutures, respectively. The aorta-graft entrance and heart-graft inflow tract were then constructed by sharing to connect the proximal anastomotic site of the recipient infrarenal aorta. Briefly, the left side of the anastomotic site of the recipient infrarenal aorta was anastomosed to the posterior wall of the donor aorta-graft inside vessel (**Fig 2B**). In the following step, the donor heart graft was placed on the left side of the recipient abdomen with the remnant of the ascending aorta underneath the pulmonary artery and perpendicular to the clamped vessels, and then covered with gauze moistened with ice-cold saline. The anterior wall of the aorta-graft was anastomosed to the posterior wall of the ascending aorta of the heart-graft inside vessel (**Fig 3A and 3B**). The anterior wall of the ascending aorta of the donor heart was then anastomosed to the right side of the anastomotic site of the recipient infrarenal aorta outside vessel (**Fig 4A and 4B**). The last step in the procedure was to construct the heart-graft outflow tract, similarly, by an end-to-side anastomosis between the donor pulmonary artery and the recipient IVC with continuous running sutures of the posterior wall inside vessel and the anterior wall outside vessel (**Fig 5A**).

After completion of anastomoses, the distal ligature was untied first, followed by removal of the proximal clamp. The anastomotic sites were gently pressed by two cotton swabs for several seconds, to minimize/stop bleeding. Optionally, small pieces of hemostatic agent, spongostan, can be placed around the anastomotic sites to prevent anastomotic bleeding. Normally, the aorta- and heart- graft immediately fills with blood and consequently becomes bright red in

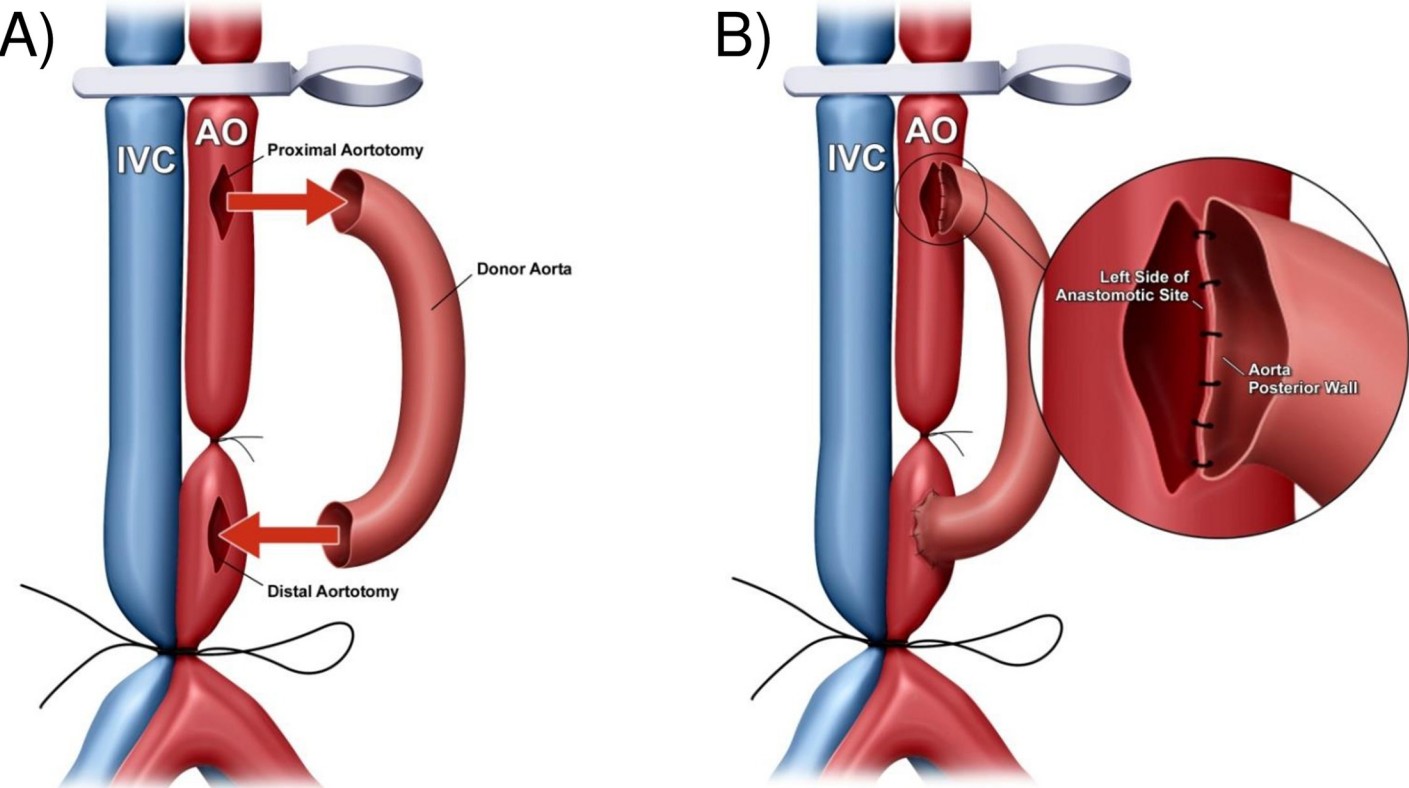

**Fig 2.** (A) Diagrammatic representation of the proximal and the distal aortotomies on the recipient's infrarenal aorta were performed above and below the ligature, respectively, for the subsequent anastomoses of the donor's aorta graft into the recipient in an end-to-side fashion. (B) Diagrammatic representation of the left side of the anastomotic site of the recipient's infrarenal aorta, which was anastomosed to the posterior wall of the donor's aorta graft within the vessel.

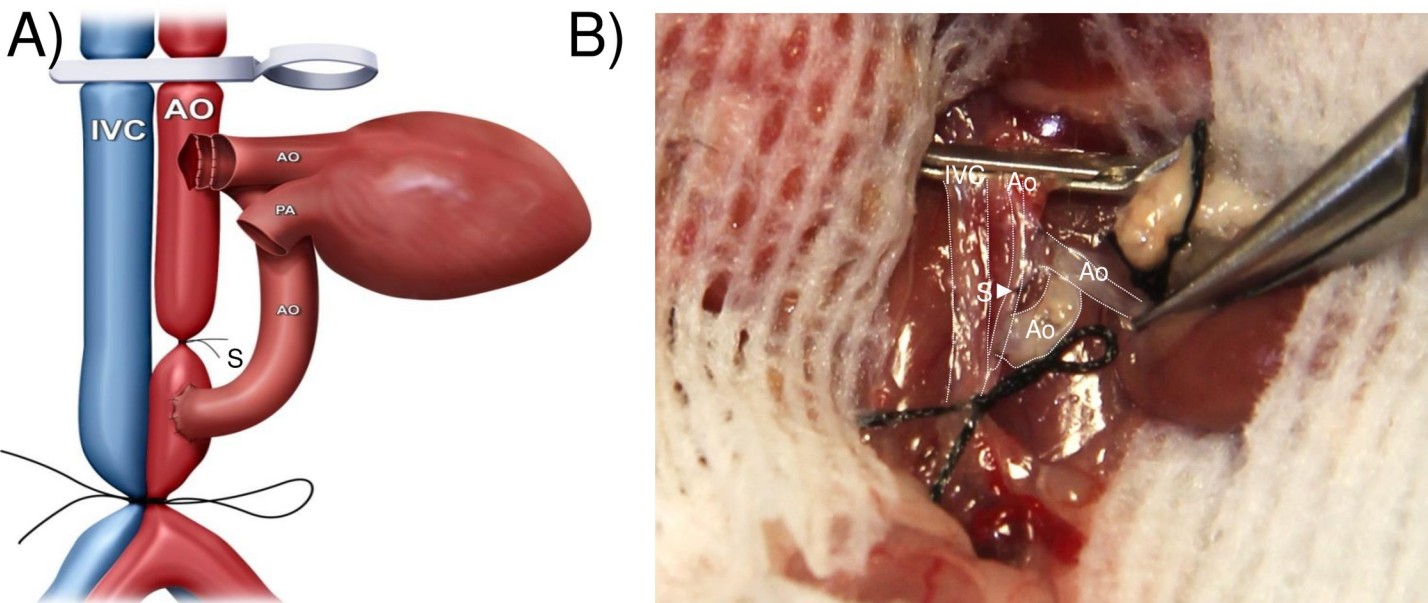

**Fig 3.** Diagrammatic (A) and photographic (B) representation of the anterior wall of the aorta graft, which was anastomosed to the posterior wall of the ascending aorta of the heart graft within the vessel.

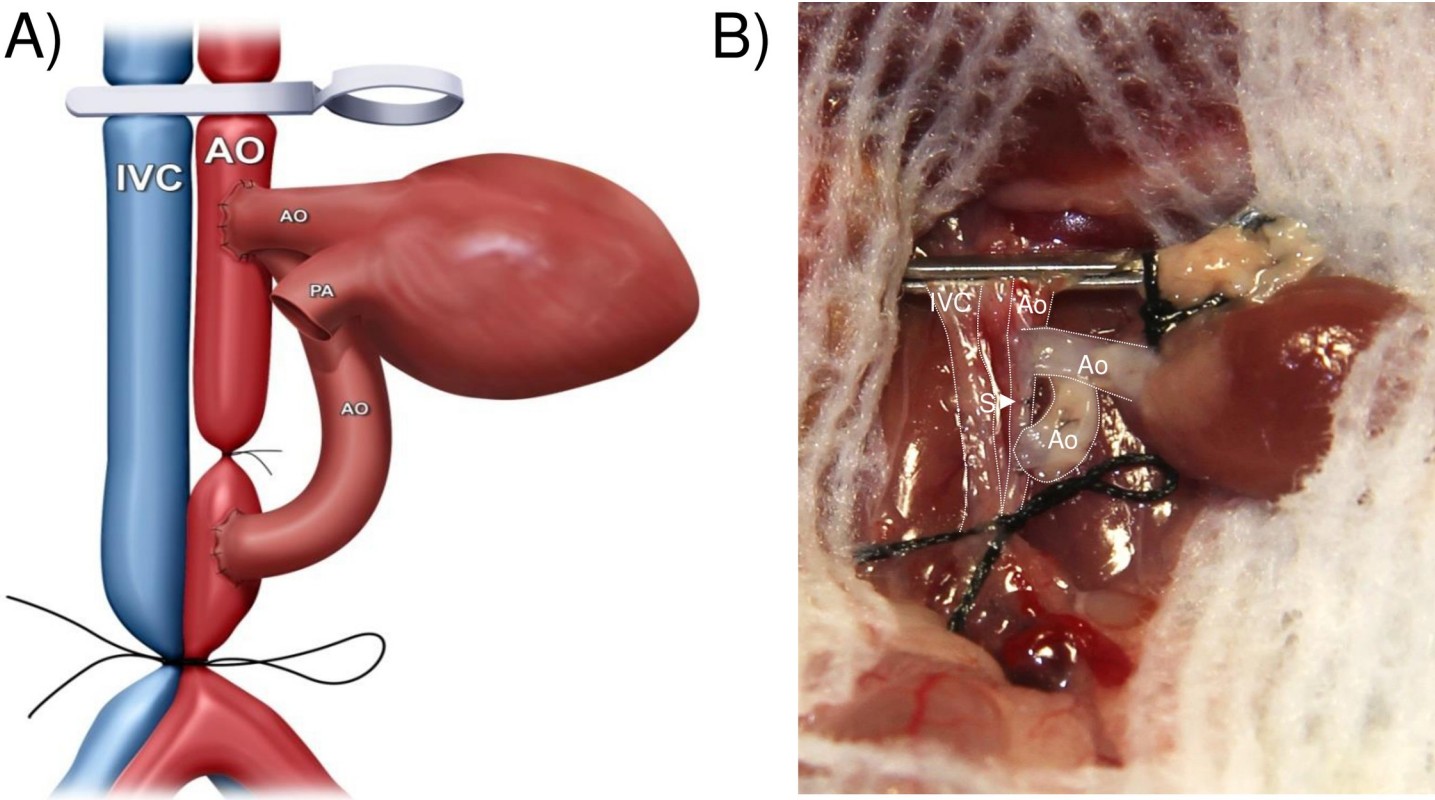

**Fig 4.** Diagrammatic (A) and photographic (B) representation of the anterior wall of the ascending aorta of the heart graft, which was anastomosed to the right side of the anastomotic site of the recipient's infrarenal aorta.

color. Prominent pulsations of the aorta-graft were visible and after a short episode of fibrillation, sinus rhythm resumed in the heart-graft (Fig 5B). After ensuring there was no further bleeding, the abdominal incision was closed with 4–0 suture by continuous running stitches.

**Immunosuppression.** For a subset of additional transplants, the long-term viability of this combined aorta and heart transplant model were tested in syngraft (C57Bl/6 to C57Bl/6) (n = 3) and allograft (Balb/c to C57Bl/6) models (n = 3), with the administration of 250 μg of *InVivoMab* anti-mouse CD40L (CD154; clone MR-1; Bio X Cell, West Lebanon, NH) on the day of transplantation, and 200 μg of *InVivoMab* recombinant CTLA-4-Ig (hum/hum; Bio X Cell) on day 2 post-transplantation [17, 18].

**Histology and imaging.** Four representative sections from each aortic and heart graft were stained with hematoxylin and eosin (H&E; Leica Biosystems, Buffalo Grove, IL) or Movat's pentachrome (Newcomer Supply, Middleton, WI) and digital images were captured using an Olympus BX61 microscope (Center Valley, PA).

## Results

A total of 30 combined heterotopic abdominal heart and aorta syngrafts were performed using this technique in *C57Bl/6* mice. A median time of 11.5 min (10–14 min range) and 8.0 min (7–10 min range) was required for harvesting of the donor heart and aorta grafts, respectively. Preparation of the transplant recipient required a median time of 9.0 min (8–11 min range), and anastomoses of the donor organs required a median time of 40.5 min (38–44 min range). There was a median total ischemic time of 70 min (65–74 min range) (Fig 6). The surgery

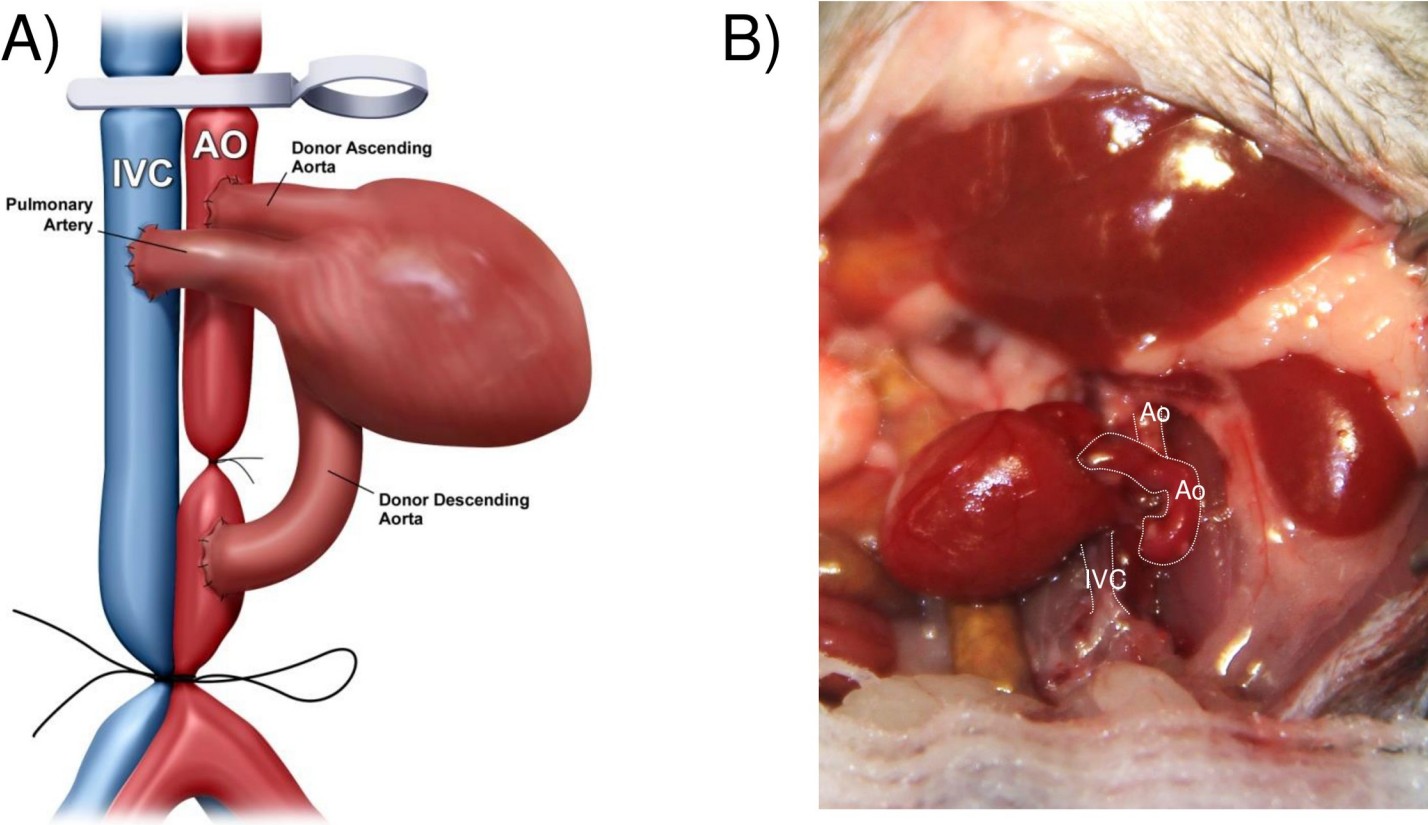

**Fig 5.** Diagrammatic (A) and photographic (B) representation of the outflow tract (pulmonary artery) of the donor's heart graft, which was connected to the recipient's inferior vena cava (IVC).

survival rate of syngrafts was greater than 96% (29/30). One mouse died due to anastomotic hemorrhage on the same day as transplantation. The remaining 29 of 30 syngrafts survived greater than 90 days with maintained transplanted heart function (as assessed by palpation).

To validate this surgical method, we subsequently performed allografts using *Balb/c* donor aorta and hearts transplanted into *C57Bl/6* recipients (n = 3) in the presence of double co-stimulatory blockade [17, 18] or syngrafts using *C57Bl/6* donor aorta and hearts transplanted into *C57Bl/6* recipients (n = 3) as a control. For this group, 3/3 allografts and 3/3 syngrafts survived for more than 90 days, with maintained transplanted heart function (as assessed by palpation). Histological examination of aorta (Fig 7A) or cardiac (Fig 7B) grafts revealed long-term survival with a mild immune response, due to the strong immunosuppressive effect of double co-stimulatory blockade.

## Discussion

AV is resistant to current immunosuppressive therapies and a major obstacle to long-term survival in solid organ transplantation. Although, there are controversies concerning the relative contribution of acute transplant rejection to the subsequent development of AV, current evidence increasingly suggests that acute rejection is a strong risk factor for the late development of AV [1, 2]. Indeed, clinical reduction in acute rejections has significantly decreased the incidence of chronic rejection graft failure [1, 2, 19].

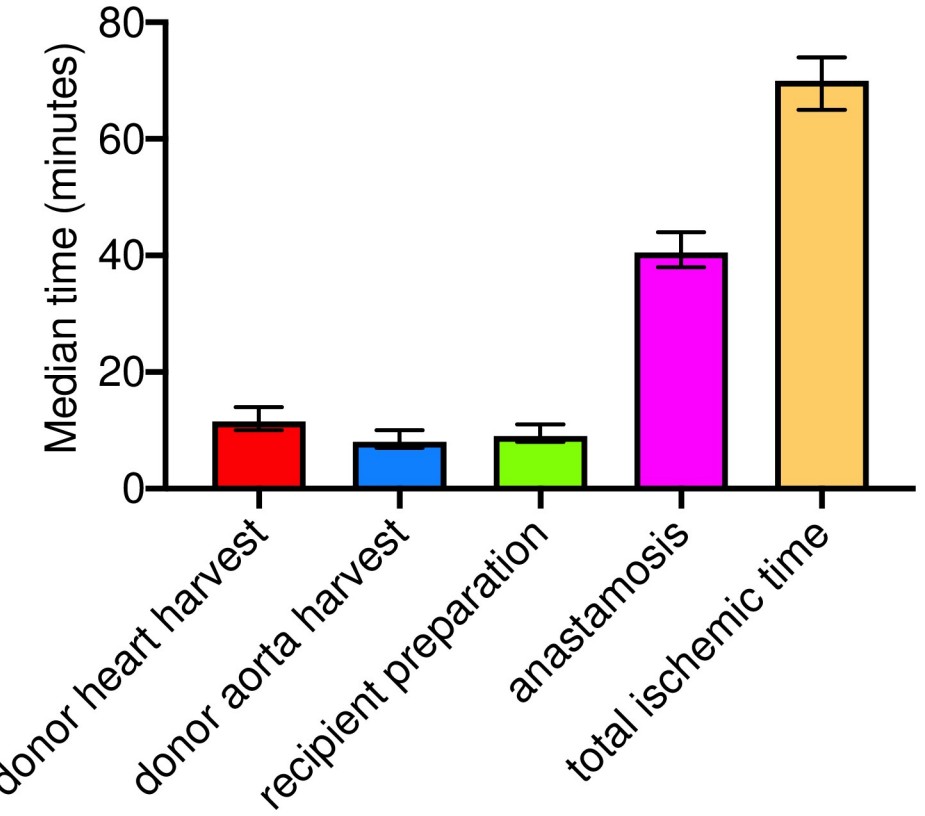

**Fig 6. Median time for donor heart harvest (red), donor aorta harvest (blue), transplant recipient preparation (green), total anastomosis time for the combined transplant procedure (pink) and total ischemic time (orange).** Data are represented as median time (in minutes) ± the range in time for all procedures. n = 30 transplants.

The aortic transplant model in mice is a useful tool for studying AV, based on the formation of consistent lesions which can be easily quantified [20]. However, unlike solid organ allografts, isolated aortic allografts can achieve long-term acceptance in the absence of immunosuppressive interventions, sparing them from the influence of acute rejection. Therefore, this model lacks the contribution of cellular and soluble factors present in acute rejection that may augment the pathogenesis and evolution of CAV. In contrast, the strategy of combining aortic transplantation with solid organ transplantation includes the influence of systemic factors related to acute allograft rejection on the development of CAV, thus providing a model that may more accurately represent chronic allograft rejection. Indeed, Soleimani *et al*. demonstrated a marked increase in neointimal area in *C57Bl/10* carotid grafts co-transplanted with *Balb/c* cardiac allografts into a *C3H* recipient mouse, compared to *C57Bl/10* carotid grafts into *C3H* recipients [14]. This observation confirmed that the acute parenchymal rejection is an important contributor to CAV, and further emphasizes that a combined model may serve as a more clinically relevant model to investigate the mechanisms involved in the development of AV.

Researchers have developed the combined aorta/carotid and heart transplantation models in mice, in which the aorta/carotid and heart are transplanted into two different surgical sites (abdomen and cervical region, respectively). A likely rationale for the use of an additional surgical site is that there is insufficient space along the recipient abdominal aorta between the renal artery and its bifurcation to make three aortotomies for connecting aorta-graft entrance and exit as well as heart-graft inflow tract. Thus, a cervical operation has to be performed to

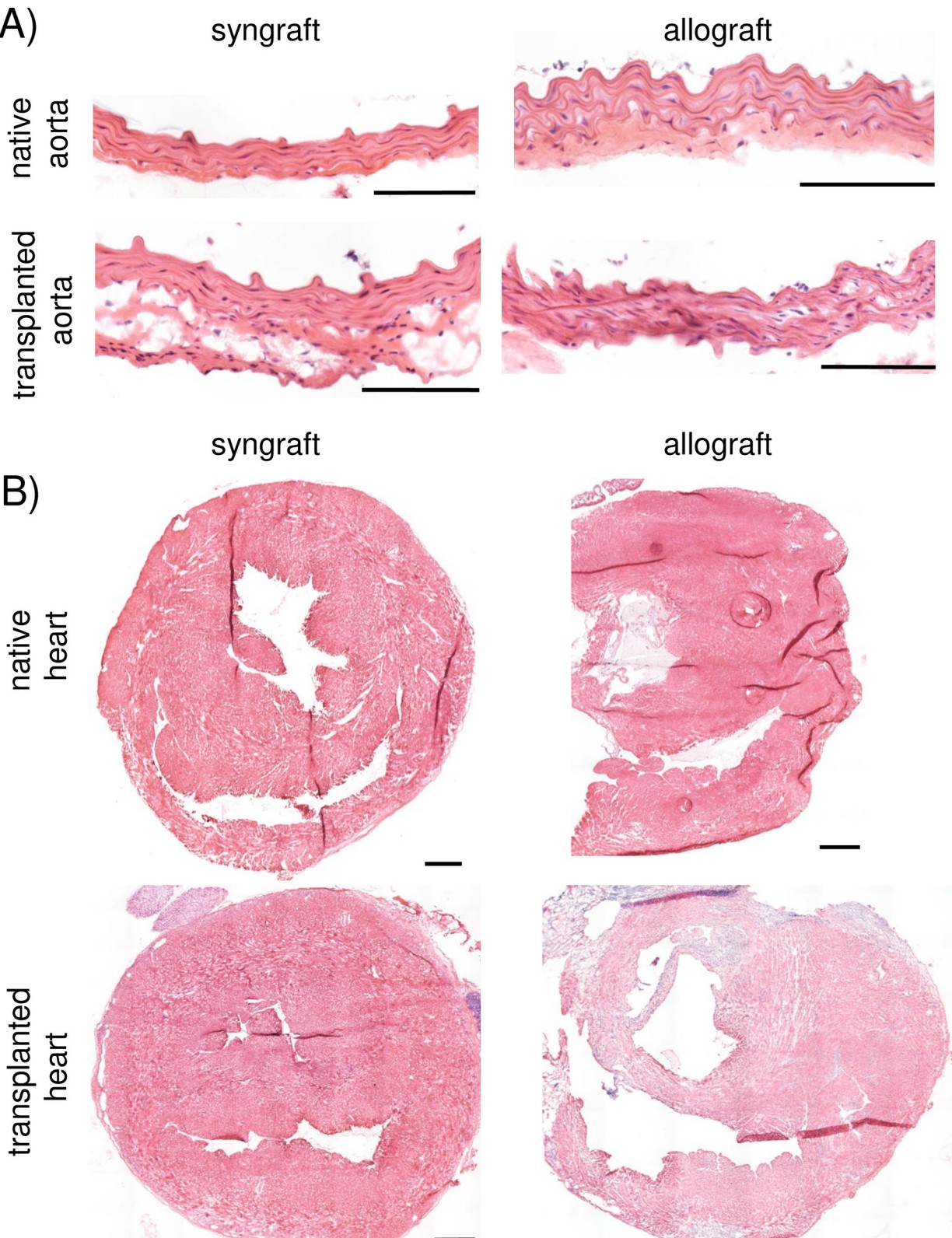

**Fig 7.** Representative micrographs of the aortic (A) and cardiac (B) grafts 90+ days post-transplantation treated with anti-CD40L and CTLA4.Ig stained with hematoxylin & eosin. Scale bars = 100 μm for (A); 500 μm for (B).

accommodate another graft. However, two surgical sites of operations are associated with additional surgical procedures and prolonged anaesthesia, frequently resulting in increased surgical complications and mortality. To avoid the unnecessary surgical traumas and shorten the ischemia and overall operative times, we developed a simplified model, in which two individual grafts can be accommodated in a single surgical site of abdomen with a high success rate.

Several arterial transplantation models have been well-established in mice [20–24]. Both aortic and carotid grafts are most commonly used for study of CAV [21, 22]. Compared to carotid segments with similar length, aortic grafts are larger in size, and therefore can provide more tissue for subsequent histological and biochemical analysis. In our experience, a whole-length of thoracic ascending aorta from a single donor is long enough to be divided and transplanted into three to four individual recipients (unpublished observation). Therefore, use of an aortic graft rather than a carotid graft is superior in such models.

In mice, aortic grafts can be anastomosed to the recipient infrarenal aorta by either an end-to-end [20] or an end-to-side [11] anastomosis pattern. Due to the obvious disparity in size between the thoracic and abdominal segments of the aorta, the end-to-end pattern is technically more difficult, with a higher incidence of anastomotic complications (up to 20% of thrombotic complications reported) [20]. In contrast, the end-to-side pattern is technically more facile, leading to a greater than 98% success rate [11]. In addition, a mild dissection of the infrarenal aorta from IVC is sufficient to perform an end-to-side anastomosis. Thus, the donor operation is simplified by obviating the dissection of abdominal vessels, which are very easily injured. Of note, a potential limitation of this pattern is that a curved loop of aortic graft may alter blood flow patterns and produce unexpected and unreliable experimental results. To address this issue, after completion of the procedure of end-to-side anastomosis, Sun *et al.* [11] tried to convert an end-to-side to a quasi end-to-end anastomosis by transection of the native infrarenal aorta between the anastomotic sites of aortic allograft entrance and exit. However, this technique appeared to increase the likelihood of kinking at the anastomotic sites [25]. Importantly, data showed a loop and an interposition of aortic allograft made by end-to-side and end-to-end pattern respectively, yielded similar experimental results [25]. Therefore, on the basis of merit, we combined the end-to-side anastomosis pattern of aortic transplant without transection of native infrarenal aorta described by Cho *et al.* [15] for our study.

We used the abdominal site to implant both heart and aortic allografts, because the abdominal infrarenal aorta is much larger in size and longer than the carotid artery. Although the infrarenal aorta was divided from the IVC, a complete separation of these vessels from each other was unnecessary. Instead, a mild dissection of the abdominal vessels was easily performed and adequate to facilitate an end-to-side anastomosis. In order to transplant both heart and aorta grafts, three anastomotic stomas in the recipient infrarenal aorta were needed for reconstruction of aorta-graft entrance and exit as well as heart-graft inflow tract. However, we discovered that two adequate aortotomies, at a maximum, can be made in recipient infrarenal aorta for anastomosis, despite attempting to replace the distal clamp with a ligature of 5–0 silk to maximize the usage of the infrarenal aorta. To address this issue, we adopted the proximal anastomotic stoma on the recipient infrarenal aorta to connect both aorta-graft entrance and heart-graft inflow tract, increasing the available space along the recipient infrarenal aorta. Results obtained from our study demonstrated this novel microsurgical technique can be easily performed with a high success rate, without thrombotic occlusion or stenosis observed in heart and aortic allografts. This novel technique also provides a new pathway in revascularization for multi-organ transplantation within a limited space.

Su *et al.* [26] argued that an entire everting suture technique was able to decrease the anastomotic complications. However, in our laboratory, we favored the parachute technique to

suture the vessel posterior walls inside the vessels (inverting suture) and anterior walls outsider the vessels (everting suture). In this way, grafts did not need to be re-positioned to the opposite side, and therefore the possibility of increased surgical complexities and warm ischemia times was further limited. In our previous experience with this method, a success rate of greater than 90% could be reliably achieved by a skilled microsurgeon for heterotopic heart transplantation in mice. In our current model, we adopted the same suture technique and achieved a similarly high success rate. In addition, using a technique described by Mao *et al.* to create the aortotomy and the venotomy [16], an elliptical anastomotic stoma can be easily made with a trim-edge, which may facilitate the vascular anastomosis. Nevertheless, proficiency in microsurgical techniques is required to enable researchers to achieve reliable and reproducible results with this model.

## Conclusions

In conclusion, our study presents a simple and reliable technique for combined heterotopic abdominal heart and aorta transplantation in mice, which can be achieved by sharing a common anastomotic stoma on the recipient infrarenal aorta with two individual graft vessels. Further studies using this model should provide additional insight into the mechanism of CAV and reveal correlation between the acute rejection and the subsequent development of CAV. In addition, the novel technique applied in this model provides a new pathway for revascularization and increases availability of methodological options for the development of experimental animal models.

## Supporting information

**S1 Data.**
(XLSX)

## Acknowledgments

We acknowledge the assistance of the Digestive Disease Research Core Center (DDRCC) at the Washington University School of Medicine for assistance with histology.

## Author Contributions

**Conceptualization:** Hao Dun.

**Data curation:** Hao Dun, Li Ye, Yuehui Zhu.

**Formal analysis:** Brian W. Wong.

**Funding acquisition:** Brian W. Wong.

**Investigation:** Hao Dun, Brian W. Wong.

**Methodology:** Hao Dun.

**Project administration:** Hao Dun, Brian W. Wong.

**Resources:** Brian W. Wong.

**Supervision:** Brian W. Wong.

**Writing – original draft:** Hao Dun.

**Writing – review & editing:** Hao Dun, Brian W. Wong.

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
