## [Decision Letter · Decision Letter 0]

1 Apr 2020

PONE-D-20-06026

Combined abdominal heterotopic heart and aorta transplant model in mice

PLOS ONE

Dear Dr. Wong,

Thank you for submitting your manuscript to PLOS ONE. After careful consideration, we feel that it has merit but does not fully meet PLOS ONE’s publication criteria as it currently stands. Therefore, we invite you to submit a revised version of the manuscript that addresses the points raised during the review process.

We would appreciate receiving your revised manuscript by May 16 2020 11:59PM. To enhance the reproducibility of your results, we recommend that if applicable you deposit your laboratory protocols in protocols.io, where a protocol can be assigned its own identifier (DOI) such that it can be cited independently in the future. For instructions see: http://journals.plos.org/plosone/s/submission-guidelines#loc-laboratory-protocols

We look forward to receiving your revised manuscript.

Kind regards,

Academic Editor

PLOS ONE

Journal Requirements:

'This work is supported, in part, by the International Society for Heart and Lung

Transplantation (ISHLT) Joel D. Cooper Career Development Award (BWW).'

Additional Editor Comments (if provided):

Please address the issues and concerns from the reviewers.

Reviewers' comments:

Reviewer's Responses to Questions

**Comments to the Author**

1. Is the manuscript technically sound, and do the data support the conclusions?

Reviewer #1: Yes

Reviewer #2: Yes

2. Has the statistical analysis been performed appropriately and rigorously? 

Reviewer #1: Yes

Reviewer #2: N/A

3. Have the authors made all data underlying the findings in their manuscript fully available?

Reviewer #1: Yes

Reviewer #2: Yes

4. Is the manuscript presented in an intelligible fashion and written in standard English?

Reviewer #1: Yes

Reviewer #2: Yes

5. Review Comments to the Author

Reviewer #1: “Combined Abdominal Heterotopic Heart and Aorta Transplant Model in Mice” is a manuscript that presents a novel microsurgical technique intended to improve upon the heart and aorta/carotid graft. The presented technique improves on the existing one because it has shorter graft ischemic times and is centered in only one body cavity. This model and the existing model are important as it may be used to recapitulate the clinical process of aortic vasculopathy. The methodology of the surgery is meticulously explained, and the statistics are appropriately performed. The illustrations are of good quality and add to the overall work. The discussion content is well thought-out and provides context for the study. This paper is marred by poor grammar, and since PLOS one does not provide proofreading and copyediting support, I do not believe it is publishable in its current state. I also suggest the following minor revisions for this manuscript in order to consider it for publication:

It would be helpful to the reader to provide the true distributions of the times for the operation (ischemic time, harvesting time), instead of reporting the approximate times. The authors should present the median time and the distribution (this could be represented graphically as well).

The validation of the new surgical method should be more adequately described. The authors write, “we subsequently performed allografts using Balb/c donor aorta and hearts transplanted into C57Bl/6 recipients in the presence of double co-stimulatory blockade”. The number of replicates of this procedure should be detailed, and it should be noted that the histologic examination of the aorta and cardiac graphs (6a and 6b) are representative of these experiments, if that is in fact the case.

Please add citations for the following sentence:

• Chronic allograft rejection remains the leading cause of graft loss one-year post-transplantation.

PLOS one does not provide any copyediting and proofreading service. There are multiple grammatical errors found in this manuscript. I believe that it is not publishable in its current state. I would suggest proofreading the article again. Examples of the many grammatical errors found throughout the paper are listed below:

• The result has further heightened the need for a parenchymal organ in combination with aorta transplantation in exploring clinic(al) relevance of CAV.

• Our results demonstrate that this new model is characterized by the reliable reproducibility.

oShould this sentence include “the”?

• After 1 minute for the systemic heparinization, an aortotomy of abdominal aorta is made to decompress the blood circulatory system,

othis should be an aortotomy of “the” abdominal aorta

• The IVC and the right superior vena cava (SVC) were proximally ligated with 7-0 sutures, respectively.

oIn this case respectively does not make sense.

Reviewer #2: Dear authors,

Thank you for submitting your manuscript titled "Combined abdominal heterotopic heart and aorta transplant model in mice" to PLOS ONE. It was a very well written manuscript and I commend you on accomplishing this very difficult microsurgery. Your description of the operation, as well as the diagrams truly help the reader follow along the operation. I do have a couple questions :

1. Have you compared this technique to the "conventional" heterotopic heart transplantation in mice? As in, have you compared the histology, alloresponses, coronary inflammation, etc between the techniques. This technique seems more difficult, with more anastomoses. If everything is similar, is there a reason to do this technique?

2. Unlike other solid organs, the heart is a dynamic structure that can fail with improper preload and afterload management which is critical post-transplantation. This can affect immunosuppression. Since this technique creates an organ in series, it may not be representative of the pressures seen in a real transplant. This may not matter in a liver for example. Do you think that since the heart is not exposed the similar pressures as in an orthotopic heart transplant that this technique may not be the best to address CAV? Please address. I think this is a major limitation of such models.

3. Despite heparin, there could still be issues with thrombosis. How can you tell if the coronaries are patent in the transplanted heart? Is the mouse continued on anticoagulation?

4. Do the transplant hearts go into arrhythmia?

5. Please review grammar and syntax.

Thank you very much for a very interesting manuscript.

6. PLOS authors have the option to publish the peer review history of their article (what does this mean?). If published, this will include your full peer review and any attached files.

Reviewer #1: No

Reviewer #2: No

---

## [Author Response · Author response to Decision Letter 0]

23 Apr 2020

Please find our response to the reviewers and editor comments in the attached 'Response to Reviewers' file.

---

## [Decision Letter · Decision Letter 1]

19 May 2020

PONE-D-20-06026R1

Combined abdominal heterotopic heart and aorta transplant model in mice

PLOS ONE

Dear Dr. Wong,

Thank you for submitting your manuscript to PLOS ONE. After careful consideration, we feel that it has merit but does not fully meet PLOS ONE’s publication criteria as it currently stands. Therefore, we invite you to submit a revised version of the manuscript that addresses the points raised during the review process.

Before we accept the manuscript, please address the issue found by one reviewer:

At the end of the first paragraph of the results the following sentence is hard to understand: “Survival of both the syngeneic aorta and heart graft was greater than 90 days.” Is this referring to all of the mice that survived the procedure?

I do not believe that the graft survival should be presented as 6/6 in the second paragraph of the results (since you are referring to three cardiac graphs and three aortic graphs). I think it should be presented as 3/3 cardiac graphs and 3/3 aortic graphs.

We would appreciate receiving your revised manuscript by Jul 03 2020 11:59PM. To enhance the reproducibility of your results, we recommend that if applicable you deposit your laboratory protocols in protocols.io, where a protocol can be assigned its own identifier (DOI) such that it can be cited independently in the future. For instructions see: http://journals.plos.org/plosone/s/submission-guidelines#loc-laboratory-protocols

We look forward to receiving your revised manuscript.

Kind regards,

Academic Editor

PLOS ONE

Reviewers' comments:

Reviewer's Responses to Questions

**Comments to the Author**

1. If the authors have adequately addressed your comments raised in a previous round of review and you feel that this manuscript is now acceptable for publication, you may indicate that here to bypass the “Comments to the Author” section, enter your conflict of interest statement in the “Confidential to Editor” section, and submit your "Accept" recommendation.

Reviewer #1: (No Response)

Reviewer #2: All comments have been addressed

Reviewer #3: All comments have been addressed

2. Is the manuscript technically sound, and do the data support the conclusions?

Reviewer #1: Yes

Reviewer #2: Yes

Reviewer #3: Yes

3. Has the statistical analysis been performed appropriately and rigorously? 

Reviewer #1: Yes

Reviewer #2: Yes

Reviewer #3: N/A

4. Have the authors made all data underlying the findings in their manuscript fully available?

Reviewer #1: Yes

Reviewer #2: Yes

Reviewer #3: Yes

5. Is the manuscript presented in an intelligible fashion and written in standard English?

Reviewer #1: Yes

Reviewer #2: Yes

Reviewer #3: Yes

6. Review Comments to the Author

Reviewer #1: The authors of Combined abdominal heterotopic heart and aorta transplant model in mice present a much improved study detailing a new microsurgical technique aimed at studying allograft vasculopathy. My prior comments on their manuscript have been adequately addressed; the grammar is much improved and the distributions of the procedure times are adequately detailed for the reader. The article’s introduction and discussion have been largely re-written, and the article is now fit for publication. Below, I have two comments for the authors to consider.

At the end of the first paragraph of the results the following sentence is hard to understand: “Survival of both the syngeneic aorta and heart graft was greater than 90 days.” Is this referring to all of the mice that survived the procedure?

I do not believe that the graft survival should be presented as 6/6 in the second paragraph of the results (since you are referring to three cardiac graphs and three aortic graphs). I think it should be presented as 3/3 cardiac graphs and 3/3 aortic graphs.

Reviewer #2: Dear authors,

Thank you for submitting your manuscript titled "Combined abdominal heterotopic heart and aorta transplant model in mice" to PLOS ONE. I would like to also thank you for the time and effort spent on the revisions. It is evident that significant portions of this manuscript has been revised. Grammatical errors have been addressed and the overall ease of reading has improved. Furthermore, you have addressed all of the comments submitted by myself and my co-reviewer in a very thorough manner. Please ensure grammar and syntax are appropriate. I would like to accept this manuscript.

Thanks

Reviewer #3: The methodology of the surgery is meticulously explained with nice schemes and photos. I have no further comment to be addressed.

7. PLOS authors have the option to publish the peer review history of their article (what does this mean?). If published, this will include your full peer review and any attached files.

Reviewer #1: No

Reviewer #2: No

Reviewer #3: No

---

## [Author Response · Author response to Decision Letter 1]

20 May 2020

Please find response to the reviewer's comments in the attached file, and also copied and pasted below:

RESPONSE TO THE REVIEWER’S COMMENTS

At the end of the first paragraph of the results the following sentence is hard to understand: “Survival of both the syngeneic aorta and heart graft was greater than 90 days.” Is this referring to all of the mice that survived the procedure?

RESPONSE: The highlighted sentence indeed refers to all of the syngrafts that survived the surgical procedure (29 of the 30 performed in this group). We have revised the sentence in the manuscript to clarify this detail as follows:

“The remaining 29 of 30 syngrafts survived greater than 90 days with maintained transplanted heart function (as assessed by palpation).”

I do not believe that the graft survival should be presented as 6/6 in the second paragraph of the results (since you are referring to three cardiac graphs and three aortic graphs). I think it should be presented as 3/3 cardiac graphs and 3/3 aortic graphs.

RESPONSE: The previous sentence was meant to refer to 6 of 6 total transplants for this experimental group (three Balb/c to C57Bl/6 allografts receiving double co-stimulatory blockade and three C57Bl/6 to C57Bl/6 syngrafts). We have revised the sentence in the manuscript to clarify this detail as follows:

“For this group, 3/3 allografts and 3/3 syngrafts survived for more than 90 days, with maintained transplanted heart function (as assessed by palpation.”

---

## [Decision Letter · Decision Letter 2]

5 Jun 2020

Combined abdominal heterotopic heart and aorta transplant model in mice

PONE-D-20-06026R2

Dear Dr. Wong,

We’re pleased to inform you that your manuscript has been judged scientifically suitable for publication and will be formally accepted for publication once it meets all outstanding technical requirements.

Kind regards,

Academic Editor

PLOS ONE

Additional Editor Comments (optional):

Reviewers' comments:

Reviewer's Responses to Questions

**Comments to the Author**

1. If the authors have adequately addressed your comments raised in a previous round of review and you feel that this manuscript is now acceptable for publication, you may indicate that here to bypass the “Comments to the Author” section, enter your conflict of interest statement in the “Confidential to Editor” section, and submit your "Accept" recommendation.

Reviewer #1: All comments have been addressed

Reviewer #3: All comments have been addressed

2. Is the manuscript technically sound, and do the data support the conclusions?

Reviewer #1: Yes

Reviewer #3: Yes

3. Has the statistical analysis been performed appropriately and rigorously? 

Reviewer #1: Yes

Reviewer #3: N/A

4. Have the authors made all data underlying the findings in their manuscript fully available?

Reviewer #1: Yes

Reviewer #3: Yes

5. Is the manuscript presented in an intelligible fashion and written in standard English?

Reviewer #1: Yes

Reviewer #3: Yes

6. Review Comments to the Author

Reviewer #1: No Further Comments. The authors have sufficiently revised the manuscript. It is now fit for publication in Plos One.

Reviewer #3: Authors revised the manuscript in accordance with previous reviewers' comments.

I have no further comment to be addressed.

7. PLOS authors have the option to publish the peer review history of their article (what does this mean?). If published, this will include your full peer review and any attached files.

Reviewer #1: No

Reviewer #3: No

---

## [Editor Report · Acceptance letter]

10 Jun 2020

PONE-D-20-06026R2 

Combined abdominal heterotopic heart and aorta transplant model in mice 

Dear Dr. Wong:

I'm pleased to inform you that your manuscript has been deemed suitable for publication in PLOS ONE. Congratulations! Your manuscript is now with our production department. 

Kind regards, 

on behalf of

Dr. Robert Jeenchen Chen 

Academic Editor

PLOS ONE